# Two Faces of Indole-3-Carbinol—Analysis of Lipid Peroxidation Induced by Fenton Reaction Substrates in Porcine Ovary and Kidney Homogenates

**DOI:** 10.3390/nu17193032

**Published:** 2025-09-23

**Authors:** Anna K. Skoczyńska, Jan Stępniak, Małgorzata Karbownik-Lewińska

**Affiliations:** 1Department of Endocrinology and Metabolic Diseases, Medical University of Lodz, 281/289 Rzgowska St., 93-338 Lodz, Poland; anna.skoczynska@umed.lodz.pl (A.K.S.); jan.steniak@umed.lodz.pl (J.S.); 2Polish Mother’s Memorial Hospital—Research Institute, 281/289 Rzgowska St., 93-338 Lodz, Poland

**Keywords:** indole-3-carbinol, porcine ovary, porcine kidney, Fenton reaction, lipid peroxidation

## Abstract

**Background**: Indole-3-carbinol is an indole derivative which is commonly present in vegetables, which belong to *Brassicaceae family* and has many medicinal properties. This study aimed to investigate the antioxidant impact of indole-3-carbinol on damages caused by Fenton reaction substrates to lipid membranes (lipid peroxidation) of porcine kidneys and ovaries. **Methods**: Antioxidant effect of indole-3-carbinol was assessed using Lipid Peroxidation Assay. As damaging agents were used Fenton reaction substrates, i.e., FeSO_4_ at 11 different concentrations and H_2_O_2_. The concentrations of indole-3-carbinol were 0.0, 20.0, 10.0, 5.0, 1.0 and 0.5 mM. **Results**: Comparative analysis showed higher lipid peroxidation levels in kidney than ovary homogenates at 600–18.75 μM FeSO_4_. Indole-3-carbinol significantly reduced LPO in porcine ovary homogenates at higher FeSO_4_ concentrations (1200–300 μM) in a concentration-dependent manner, while antioxidant effects in kidney homogenates were observed across a broader FeSO_4_ range (1200–18.75 μM). Notably, at the lowest FeSO_4_ concentrations (4.687–2.343 μM), high doses of indole-3-carbinol (20.0 and 10.0 mM) induced pro-oxidant effects in both tissues. Furthermore, indole-3-carbinol at these concentrations exhibited potential pro-oxidant activity even in samples without added Fenton reaction substrates. **Conclusions**: Indole-3-carbinol has dose-dependent antioxidant effects in porcine ovary and kidney homogenates under high oxidative stress, reducing Fenton reaction-induced lipid peroxidation. However, high doses of indole-3-carbinol exhibited pro-oxidant effects at lower prooxidant concentration and under basal conditions (i.e., without addition of prooxidant), highlighting the importance of dose and oxidative conditions in its potential therapeutic use.

## 1. Introduction

Reactive Oxygen Species (ROS) are chemical molecules, which level is varied depending on the physiological state of cells. This is due to the fact, that they regulate cellular homeostasis and primarily contribute to their dysfunction. Such changes promote the development of diseases that are caused by abnormal cellular metabolism and the initiation of the inflammatory process, alteration of gene expression of inflammatory cytokines, chemokines and growth factors [1]. The change in the balance between antioxidants and free radicals contributes to different diseases such as heart attack, diabetes, atherosclerosis or tumours [1,2,3,4]. Metabolic disorders cause reduction of antioxidant system potential, that is expressed by the reduced activities of enzymes such as superoxide dismutase (SOD), glutathione peroxidase (GPx), or catalase (CAT). Besides, the presence of ROS leads to oxidative damage to proteins, lipids, and DNA, which can result in aging or cancer initiation.

Indole-3-carbinol is a known indole derivative, which naturally occurs in cruciferous vegetables from *Brassicaceae* family, such as cauliflower, broccoli and brussels sprouts [5]. Also different sorts of cabbage contain this indole derivative [6]. Vegetables from this plant family contain a lot of glucobrassicin and neoglucobrassicin [7], which are glucosylates. Myrosinase is an enzyme that hydrolyzes glucobrassicin and also is found in these vegetables. This enzyme is released from cells in mechanical processes, e.g., during chewing in the oral cavity or during food preparation. Hydrolysis of glucobrassicin leads to the formation of indole-3-carbinol. Oligomerization of indole-3-carbinol in acidic environment leads to receiving diindolylmethane [8]. In vitro studies have shown that compounds found in *Brassicacae* plants are most effective in preventing breast, cervical and endometrial cancers [9]. The anti-cancer properties of these compounds result from their influence on the cell cycle, repair of damaged DNA, apoptosis of cancer cells and limiting the proliferation of cancer cells [10]. In addition, these compounds demonstrate the ability to affect the estrogens and inhibit inflammatory processes. Antitumor activity of indole-3-carbinol relies on cell cycle stopping in G1 phase, apoptosis induction, inhibition of angiogenesis and signal transduction in cells. Activation of mitochondrial pathway through releasing of cytochrome c and stimulation of caspases’ cascade with deactivation of pathways such as PI3K/AKT, MAPK, Bcl-2 and NF-κB constitute a complex mechanism of molecular anticancer activity of indole-3-carbinol [11]. Furthermore, indole-3-carbinol prevents damages caused by ROS and Reactive Nitrogen Species (RNS) and that is a reason to recognize it as a cardioprotective agent [12]. It is worth mentioning, that indole-3-carbinol and its oligomers are able to affect activity of biotransformation enzymes, which are involved in neutralization of compounds that pose a threat to human health, such as drugs, toxins, steroid hormones, or carcinogens. For example, indole-3-carbinol can inhibit cytochrome enzymes (mainly CYP1B1 and CYP19) that participate in biosynthesis of estrogen and steroid hormones [13,14]. Other interesting experiments have shown that indole-3-carbinol inhibited lipid peroxidation caused by carbon tetrachloride in the microsomal system [10].

In the current research article, we present results of lipid peroxidation in porcine kidney and ovary homogenates. We consider indole-3-carbinol as a potential antioxidant agent, protecting against experimentally induced oxidative damage. In our study FeSO_4_ and H_2_O_2_, being substrates of Fenton reaction, were used as pro-oxidants and this is a well-known model to induce oxidative damage to membrane lipids (lipid peroxidation). It should be stressed that indole-3-carbinol is a dietary supplement. Additionally, we decided to check antioxidant and pro-oxidant effects of indole-3-carbinol, because literature data do not refer to any studies that would concern such activity of indole-3-carbinol.

## 2. Materials and Methods

### 2.1. Reagents

Indole-3-carbinol, FeSO_4_ and H_2_O_2_ were obtained from Sigma (St. Louis, MO, USA), and concentrated C_2_H_5_OH was obtained from Stanlab (Lublin, Poland). For investigation of lipid peroxidation the LPO-586 kit was used from Enzo Life Science (Farmingdale, NY, USA). All chemicals have high level of purity. The concentrations of FeSO_4_ were obtained through the dilution method with use of Tris buffer (pH 7.4) and were equal to 0.0, 1200, 600, 300, 150, 75, 37.5, 18.75, 9.375, 4.687, and 2.343 µM. Concentrations of indole-3-carbinol dissolved in C_2_H_5_OH were 0.0, 20.0, 10.0, 5.0, 1.0 and 0.5 mM. The final concentration of C_2_H_5_OH during incubation was 1% (*v*/*v*). Hydrogen peroxide (H_2_O_2_) was used in concentration equal to 5 mM. Each experiment was performed three times.

### 2.2. Porcine Tissue Homogenates Preparation

According to The Polish Act on the Protection of Animals Used for Scientific or Educational Purpose and Directive 2010/63/EU of the European Parliament and the Council (published on 22 September 2010) it is not necessary to have an approval of the local ethics committee. Kidneys and ovaries tissues were obtained from pigs (aged 8–9 months) at a slaughterhouse. The animals were in good health. The tissues were immediately frozen on solid CO_2_ and stored at −80 °C.

### 2.3. Performance of Lipid Peroxidation (LPO) Assay

For homogenization of porcine tissues a Tris buffer (pH 7.4) was used and the reagents (FeSO_4_, H_2_O_2_, indole-3-carbinol) were added. Thereafter, homogenates were kept in the presence of added substances at 37 °C for 30 min.

The LPO index means the level of MDA + 4–HDA (Malondialdehyde + 4-Hydroxyalkenals) and is expressed as nmol per mg of protein. The LPO-586 kit (Enzo Life Science, Farmingdale, NY, USA) was applied as a protocol in the whole described procedure. To obtain supernatant, the samples were centrifuged (5000× *g*, 10 min, 4° C) after incubation of homogenates with the examined substances. Thereafter, the experiment was performed in duplicate with the use of supernatant and the samples were prepared as follows: 200 μL of sample supernatant was combined with 650 μL of mixture of CH_3_OH and C_2_H_3_N (prepared in 1:3 ratio) containing N-methyl-2-phenylindole. Then, 150 μL of methanesulfonic acid (15.4 M) was added to each sample and the incubation was performed at 45 °C for 40 min. After centrifugation (5000× *g*, 10 min, 4 °C), an absorbance of each sample was measured at 586 nm. An absorbance of protein samples was measured at 595 nm. Results are expressed as MDA + 4–HDA per 1 mg of protein.

### 2.4. Statistical Calculations

Statistical calculations were performed using *SigmaPlot 11.0* based on data from three independent experimental replicates.

For comparisons involving more than two groups, a one-way analysis of variance (ANOVA) was performed, followed by the Student–Neuman–Keuls’ post hoc test. For pairwise comparisons (e.g., between ovary and kidney samples), unpaired t-tests were applied. Prior to applying ANOVA or t-tests, data were tested for normality using the Shapiro–Wilk test, and for homogeneity of variances using the Levene’s test. Cohen’s d effect sizes were calculated to quantify the magnitude of differences between selected treatment groups.

Statistical significance was considered at *p* < 0.05. All results are presented as means ± SE.

## 3. Results

Here, we would like to present results of antioxidant activity of indole-3-carbinol in porcine ovary and kidney homogenates. We compared lipid peroxidation level in the both of porcine tissues homogenates in relation to control sample and concentrations of FeSO_4_ (expressed as Fe^2+^) in each tissue homogenates. As shown in Figure 1, statistically significant increases in LPO were observed compared to control at Fe^2+^ concentrations between 1200 and 75 μM in ovary homogenates (*p* < 0.001–0.006; Cohen’s d range = 4.56–17.13) and between 1200 and 18.75 μM in kidney homogenates (*p* < 0.001; Cohen’s d = 6.51–27.13). Moreover, the comparison between the two tissue types revealed statistically significant differences in LPO levels at FeSO_4_ concentrations between 600 and 18.75 μM (*p* < 0.001; Cohen’s d = 6.56–15.25), with higher LPO levels consistently detected in kidney homogenates compared to ovary homogenates.

Figure 2 presents a comparative analysis of the effects of indole-3-carbinol on lipid peroxidation in homogenates of porcine ovary and kidney tissues. Statistically significant increases in LPO were observed in kidney homogenates at indole-3-carbinol concentrations of 20–10 mM (*p* = 0.02–0.023; Cohen’s d = 2.30–4.84), and in ovary homogenates at 20 mM (*p* = 0.015; Cohen’s d = 2.31), compared to their respective controls.

Such results may suggest that indole-3-carbinol, when used in higher concentrations, reveals potential pro-oxidant effects.

In porcine ovary homogenates, at the highest FeSO_4_ concentration (1200 μM), indole-3-carbinol significantly reduced lipid peroxidation across all tested concentrations (*p* < 0.01–0.037; Cohen’s d = 1.18–4.0), indicating a robust antioxidant potential under severe oxidative stress.

At 600 μM FeSO_4_, significant reductions were observed only at 20.0 mM (*p* < 0.01; d = 6.78) and 10.0 mM (*p* < 0.01; d = 4.28), suggesting a concentration-dependent effect.

At 300 μM FeSO_4_, antioxidant effects were broader and observed at 20.0 mM, 10.0 mM, 5.0 mM, and 1.0 mM of indole-3-carbinol (*p* < 0.01–0.015 for all; Cohen’s d = 1.58–6.98).

Notably, no antioxidant activity was detected at 150 μM FeSO_4_. At intermediate FeSO_4_ levels (75 and 37.5 μM), the antioxidant effect of indole-3-carbinol was observed only at selected concentrations. Specifically, at 75 μM FeSO_4_, significant reductions in lipid peroxidation (*p* = 0.014–0.02, Cohen’s d = 2.06–2.16) were recorded at 20.0, 10.0, and 5.0 mM of indole-3-carbinol. At 37.5 μM FeSO_4_, the antioxidant activity was confined to the lowest tested concentrations (1.0 and 0.5 mM) where statistically significant decreases in lipid peroxidation were also observed (*p* = 0.042–0.048, Cohen’s d = 1.93–2.15). Interestingly, at the lowest FeSO_4_ concentrations tested (18.75–2.343 μM), indole-3-carbinol at its highest concentrations (20.0 and 10.0 mM) demonstrated a pro-oxidant effect, significantly increasing lipid peroxidation relative to controls (*p* = 0.02–0.038, Cohen’s d = 1.86–2.11). All results for porcine ovary homogenates are presented in Figure 3.

In porcine kidney homogenates (Figure 4), at FeSO_4_ concentrations of 1200, 600, 300, 150, 75, 37.5, and 18.75 μM, indole-3-carbinol generally demonstrated significant antioxidant effects at 20.0, 10.0, and 5.0 mM (*p* < 0.001–0.021). The magnitude of these effects was substantial, with Cohen’s d ranging from 4.08 to 18.89, indicating large effect sizes and strong biological relevance.

Notably, at the lowest FeSO_4_ concentrations tested (4.687 and 2.343 μM), indole-3-carbinol at its highest concentrations (20.0 and 10.0 mM) exhibited a pro-oxidant effect, significantly increasing lipid peroxidation relative to the control (*p* < 0.01–0.023; Cohen’s d = 3.79–5.21). This pattern mirrors the response previously observed in porcine ovary homogenates, underscoring the dose-dependent dual role of indole-3-carbinol as both an antioxidant and a pro-oxidant depending on the oxidative context. Moreover, observation of indole-3-carbinol’s dual effect is highlighted according to the data obtained for samples containing this indole derivative, but without FeSO_4_ and H_2_O_2_ (Figure 2).

## 4. Discussion

Indole-3-carbinol is a known dietary supplement. In general it is naturally present in the cruciferous plants. The studies of World Health Organization (WHO) indicate that consuming cruciferous vegetables activates enzymes involved in detoxification processes. This compound has the ability to convert female hormones (estrogens) into safe derivatives. It is worth mentioning that indole-3-carbinol administered orally impacted human estrogen metabolism and showed protective effect against breast cancer [14]. Affecting the reduction of estrogen activity, indole-3-carbinol can contribute to improving body composition. Supplements containing indole-3-carbinol are recommended for use by physically and mentally active people, especially athletes in figure and strength disciplines, as well as endurance and speed-strength disciplines such as combat sports. Indole-3-carbinol was very fast absorbed and distributed in the cells of kidney, liver, heart, plasma, brain and lung of mice dosed with 250 mg/kg of this phytochemical and the highest concentrations of this compound were found in the kidney and liver [14]. The supplement can also be used as an element in cancer prevention. As it was already mentioned in the Introduction section, the anti-tumor mechanisms of action are based on NF-κB, Wnt, Akt, AhR, PI3K/Akt/mTOR signaling [11]. It was also found that indole-3-carbinol at concentration of 10 μM reduced growth of cervical cancer cell lines, such as MCF-7, MDA-MB-231, and HeLa, but in case of hepatocellular cell line (HepG2) the sensitivity occurred at 5 μM concentration of indole-3-carbinol [9].

The estimated intake of indole-3-carbinol by Americans from cruciferous plants is lower than 2.6 mg/day. The intake by British people is 0.1 mg/kg of body mass. Clinical trials with indole-3-carbinol revealed that its recommended dose for a 70-kg person should be from 2.9 to 5.7 mg/kg. Indole-3-carbinol was approved by Food and Drug Administration (FDA) as a dietary supplement [14]. In Europe, indole-3-carbinol has not been allowed as a nutritional supplement. The European Commission considers it for including as a novel food, but a safety assessment is required in accordance with the new regulations for dietary supplements to allow them to be sold on the European Union (EU) market [14,15]. Moreover, indole-3-carbinol possesses many biological activities, such as antiinflammatory, antioxidant, antihypertensive, and antiarrhythmic, which are attractive for medicinal studies. Regarding antitumor properties of indole-3-carbinol, the following observations are worth mentioning. This indole substance was found to diminish expression of HPV oncogenes [16]. It was shown that application of indole-3-carbinol as a dietary supplement in a daily dose of 400 mg, equivalent to approximately one third of a head of cabbage (or 300–500 g of raw cabbage or brussels sprouts per day), reversed stage II and III cervical dysplasia, which are precancerous changes in women. This antiviral property was confirmed using the real-time RT-PCR method [17]. It was also found that indole-3-carbinol can diminish the growth of tumor in patients with recurrent laryngeal papillomatosis [18]. Besides, in in vivo conditions, studies using aged female mice have shown that indole-3-carbinol exhibits antioxidant activity dependent on the activation of Nrf2 and HO-1 pathways. These findings suggest that this phytochemical may help delay ovarian aging and mitigate damage caused by aging related processes [19]. Other interesting properties of indole-3-carbinol are anti-fibrotic and anti-migratory effects on uterine leiomyomas. Studies performed on primary myometrial and leiomyoma cells, revealed that indole-3-carbinol at a concentration of 250 µg/mL effectively reduced the proliferation of myometrial cells [20,21].

In the present study we decided to choose porcine ovary and kidney tissues due to important considerations. The ovary is an endocrine gland responsible for secretion of estrogen, progesterone, relaxin, and androgens. Many scientific publications concern the activity of indole-3-carbinol as a compound which influences estrogen metabolism, as it was mentioned before [14]. The kidney is not an endocrine gland and this tissue was selected, because it was found that indole-3-carbinol showed nephroprotective effect against toxic effects of cisplatin in the experiment performed on rats. Additionally, it was shown that indole-3-carbinol decreased the level of lipid peroxidation and elevated level of superoxide dismutase and reduced glutathione in kidney tissues of these rats [22].

Our study adds further evidence for the antioxidant potential of indole-3-carbinol. In porcine ovary homogenates, indole-3-carbinol demonstrated clear, concentration-dependent antioxidant effects, significantly reducing lipid peroxidation at higher FeSO_4_ concentrations (1200, 600, and 300 μM), while efficacy diminished at intermediate and low iron levels. Similar patterns were observed in porcine kidney homogenates, where broad antioxidant activity was seen across most FeSO_4_ concentrations (1200–18.75 μM) at 20.0, 10.0, and 5.0 mM indole-3-carbinol). These results suggest that indole-3-carbinol can act as an effective antioxidant under conditions of high oxidative stress. Referring to our previous studies with the same experimental model, i.e., with application of Fenton reaction substrates as damaging agents, but in porcine thyroid homogenates, another indole derivative—indole-3-butyric acid, which is classified as a plant growth regulator, proved to be a potential protective factor [23].

On the other hand, we also discovered that indole-3-carbinol can exhibit pro-oxidant effects under certain conditions. In both porcine ovary and kidney homogenates, pro-oxidant activity emerged at the lowest FeSO_4_ concentrations (4.687 and 2.343 μM) when the highest doses of indole-3-carbinol (20.0 and 10.0 mM) were applied. These findings suggest that its redox behavior may shift from antioxidant to pro-oxidant activity under low iron-catalyzed oxidative conditions, highlighting the need for careful consideration of dose and physiological context in potential therapeutic applications.

Also under basal conditions, i.e., without addition of Fenton reaction substrates, indole-3-carbinol induced by itself lipid peroxidation in our model, which supports above considerations. These observations are consistent with existing literature. For example, administration of 10 μmol/g diet of indole-3-carbinol to immunodeficient male mice caused harmful effects on the gastrointestinal tract, underscoring the need for caution in supplementation, especially for patients with compromised immunity such as those with cancer, AIDS, or post-transplant status [24].

Furthermore, indole-3-carbinol is known to generate free radicals under certain conditions. It has been shown to produce intracellular hydroxyl radicals in *Candida albicans* cells, supporting the possibility that at high concentrations or in specific environments, it may act as a pro-oxidant rather than an antioxidant [25].

Taken together, these findings suggest that indole-3-carbinol may be considered a bioactive compound with promising antioxidant properties for the treatment of diseases associated with an imbalance between reactive oxygen species and antioxidant defence. As such, it is worth pursuing additional research using in vivo models, for example in metabolic or cancer diseases, to better understand its detailed effects. However, further well-designed clinical studies are still needed to provide clear and definitive evidence of its impact on various human health conditions [26].

In this work, high concentrations of the plant-derived compound indole-3-carbinol were used based on a review of scientific articles on animal studies, which guided our choice of concentration values. Our experiments were conducted under in vitro conditions. For context, we refer to the reported LD_50_ values of 1410 mg/kg and 1759 mg/kg for male and female mice administered intragastrically, respectively [27]. The range of indole-3-carbinol concentrations tested was broad enough to address whether this plant compound, in addition to its antioxidant activity, could also exhibit toxic effects. We confirmed that indole-3-carbinol can generate reactive oxygen species.

Additionally we have also found that lipid peroxidation of the cell membrane was stimulated with a stronger effect in porcine kidney homogenates compared to porcine ovary homogenates. Such a difference was expected and is in agreement with our results from previous studies [28,29]. Regarding the potential mechanisms of differences in lipid peroxidation induced by Fenton reaction substrates in kidney and ovary tissue, the most probable is associated with kidney cleansing function contributing to stronger oxidative stress.

In conclusion, our findings demonstrate that indole-3-carbinol exhibits clear, dose dependent antioxidant effects in both porcine ovary and kidney homogenates, particularly under conditions of high oxidative stress induced by elevated FeSO_4_ concentrations. The consistent reduction in lipid peroxidation at higher FeSO_4_ levels suggests that indole-3-carbinol may effectively mitigate Fenton reaction–mediated oxidative damage. However, its efficacy varied with both iron and indole-3-carbinol concentrations, highlighting the importance of optimizing dosing strategies. Notably, in the absence of exogenous iron or at very low FeSO_4_ concentrations, a paradoxical pro-oxidant effect was observed at the highest indole-3-carbinol doses in both tissue types, suggesting a context-dependent redox behavior. These results underscore the need for careful consideration of both oxidative conditions and supplement dosage when evaluating the potential therapeutic applications of indole-3-carbinol as an antioxidant agent.

## Figures and Tables

**Figure 1 nutrients-17-03032-f001:**
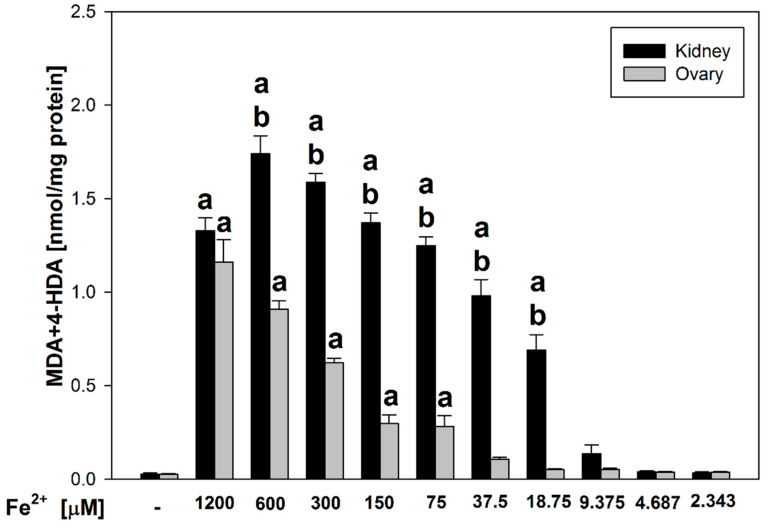
Comparison of MDA + 4–HDA concentration in porcine kidney and ovary homogenates with FeSO_4_ and H_2_O_2_ (without indole-3-carbinol). **a**
*p* < 0.05 vs. respective control ‘0’; **b*** p* < 0.05 vs. respective concentration of FeSO_4_ in ovary.

**Figure 2 nutrients-17-03032-f002:**
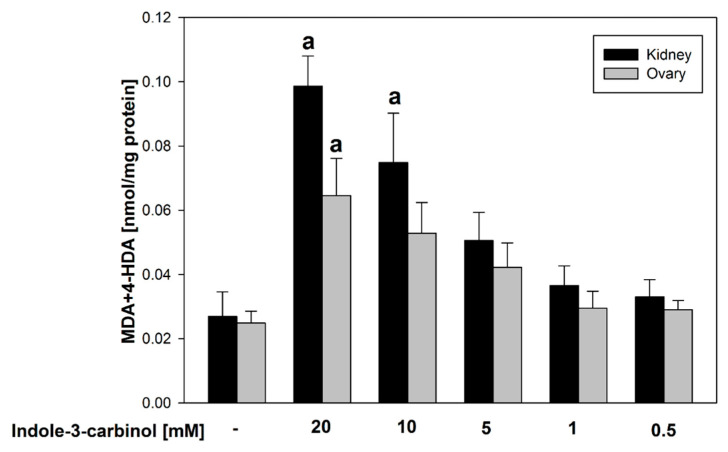
Comparison of MDA + 4–HDA concentration in porcine kidney and ovary homogenates with indole-3-carbinol (without FeSO_4_ and H_2_O_2_). **a**
*p* < 0.05 vs. respective control ‘0’.

**Figure 3 nutrients-17-03032-f003:**
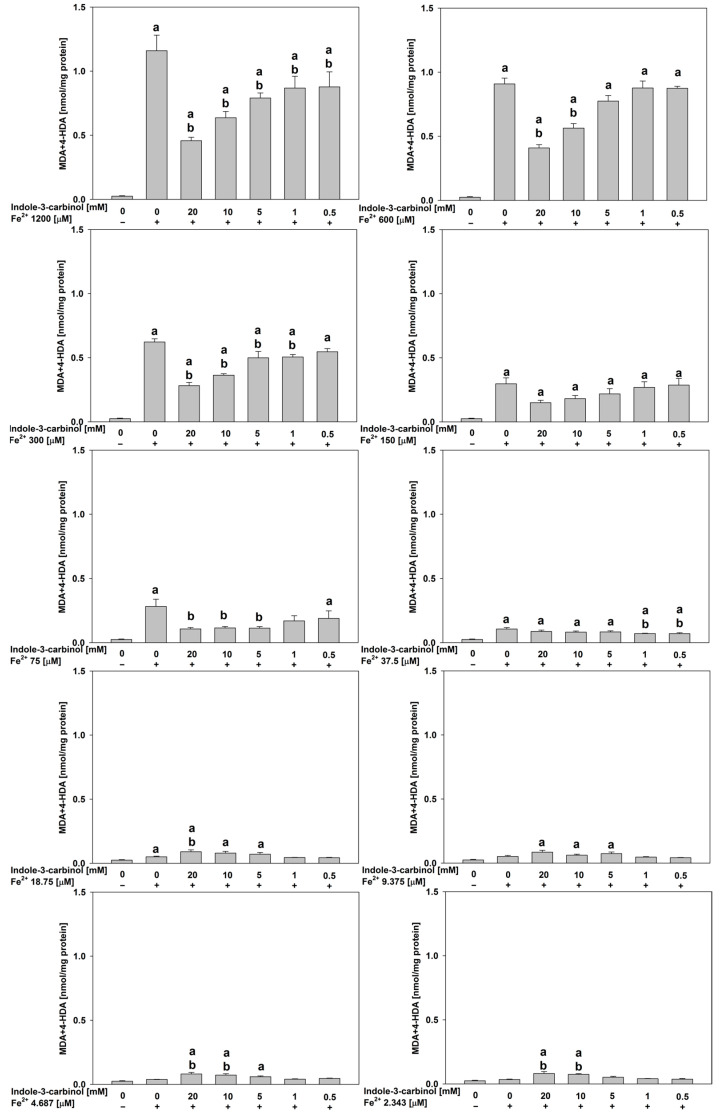
Concentration of MDA + 4–HDA in the porcine ovary homogenates with FeSO_4_, H_2_O_2_ and indole-3-carbinol. **a*** p* < 0.05 vs. control ‘0’; **b*** p* < 0.05 vs. respective concentration of FeSO_4_ with addition of H_2_O_2_ (without indole-3-carbinol).

**Figure 4 nutrients-17-03032-f004:**
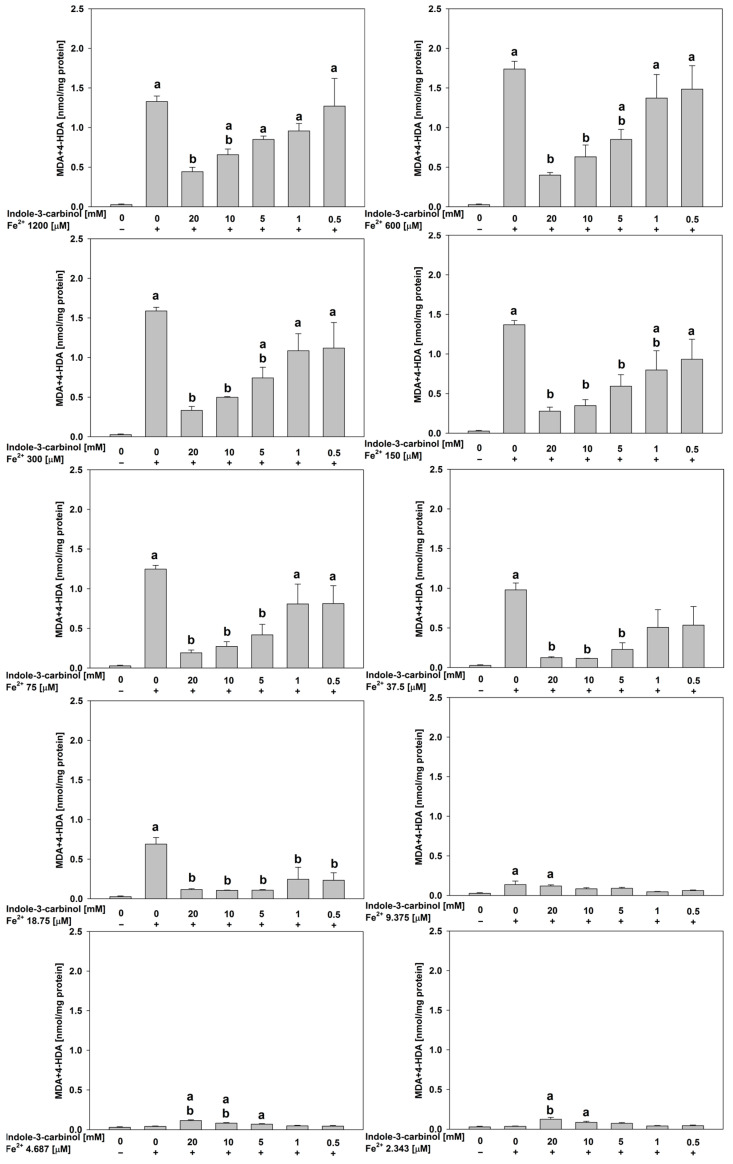
Concentration of MDA + 4–HDA in the porcine kidney homogenates with FeSO_4_, H_2_O_2_ and indole-3-carbinol. **a*** p* < 0.05 vs. control ‘0’; **b*** p* < 0.05 vs. respective concentration of FeSO_4_ with addition of H_2_O_2_ (without indole-3-carbinol).

## Data Availability

The raw data supporting the conclusions of this article will be made available by the authors on request due to privacy.

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
