# Peer review of "Two Faces of Indole-3-Carbinol—Analysis of Lipid Peroxidation Induced by Fenton Reaction Substrates in Porcine Ovary and Kidney Homogenates"

_nutrients, 2025, doi:10.3390/nu17193032_

Round 1
Reviewer 1 Report
Comments and Suggestions for Authors
The manuscript has the study is of interest, but several key methodological details, statistical reporting, ethical considerations, and clarity issues need to be addressed before it can be accepted. I recommend a moderate revision to improve these aspects and strengthen the manuscript..
Point-by-point Comments and Suggestions
- Title clarity and punctuation: The manuscript title should be clearer and include proper punctuation to make it easier to understand and more precise.
- Abstract conciseness: The abstract contains excessive detail about specific FeSO₄ concentrations. Summarizing the concentration range would enhance reader focus and overall readability.
- Research gap: The introduction does not clearly justify the choice of porcine kidney and ovary tissues nor explain how these findings translate to potential human health applications. Additionally, the manuscript does not state the specific knowledge gap regarding indole-3-carbinol’s antioxidant and pro-oxidant effects that this study aims to address.
- Methods detail: The methods section lacks information on the number of biological replicates and does not clarify how final reagent concentrations were achieved, limiting reproducibility and transparency.
- Statistical analysis: The description of statistical methods is insufficient. It should specify which tests were applied to each dataset and indicate whether assumptions underlying ANOVA and t-tests (e.g., normality, variance homogeneity) were verified.
- Results presentation: The results section is overly dense with repeated listings of concentrations and detailed data, which makes it difficult to follow. Focusing on key trends and summarizing findings more concisely would improve clarity.
- Statistical reporting: Statistical significance is frequently stated, but exact p-values, confidence intervals, or effect sizes are not provided. Including these metrics would strengthen the robustness and interpretability of the findings.
- Discussion of pro-oxidant activity: The biological implications and potential mechanisms behind the observed pro-oxidant effects of indole-3-carbinol at low FeSO₄ concentrations are not adequately explored or discussed.
- Consistency in terminology: Throughout the manuscript, there are inconsistencies in terminology and notation (e.g., FeSO₄ vs Fe²⁺), which could confuse readers. Consistent usage should be maintained.
- Several sentences are overly complex or contain minor grammatical errors, which occasionally reduce clarity. A thorough language edit would enhance readability.
- Ethical considerations: The manuscript does not include any statement regarding ethical approval or adherence to animal use guidelines for the collection of porcine tissues, which is important even when using post-mortem sample
Overall, addressing these points will improve the manuscript’s clarity, methodological rigor, and contribution to the literature on sugar intake reduction strategies.
Author Response
The manuscript has the study is of interest, but several key methodological details, statistical reporting, ethical considerations, and clarity issues need to be addressed before it can be accepted. I recommend a moderate revision to improve these aspects and strengthen the manuscript..
Point-by-point Comments and Suggestions
Comment 1: Title clarity and punctuation: The manuscript title should be clearer and include proper punctuation to make it easier to understand and more precise.
Response 1: We thank the Reviewer for this comment.
We have removed a fragment „a known dietary supplement” from the title: „Two faces of indole-3-carbinol – a known dietary supplement; analysis of lipid peroxidation induced by Fenton reaction substrates in porcine ovary and kidney homogenates”.
The title in the current form is as follows: ” Two faces of indole-3-carbinol – analysis of lipid peroxidation induced by Fenton reaction substrates in porcine ovary and kidney homogenates”.
Comment 2: Abstract conciseness: The abstract contains excessive detail about specific FeSO₄ concentrations. Summarizing the concentration range would enhance reader focus and overall readability.
Response 2: We would like to thank the Reviewer for this comment. We have removed numerical values of FeSO4 and H2O2 from an abstract. We have changed the sentence in lines 16-17. The sentence is: “As damaging agents were used FeSO4 at 11 different concentrations and H2O2, which are Fenton reaction substrates.”
Comment 3: Research gap: The introduction does not clearly justify the choice of porcine kidney and ovary tissues nor explain how these findings translate to potential human health applications. Additionally, the manuscript does not state the specific knowledge gap regarding indole-3-carbinol’s antioxidant and pro-oxidant effects that this study aims to address.
Response 3: We would like to thank the Reviewer for this comment. We explained the reasons of the choice of porcine kidney and ovary tissues in the Discussion section (lines 246-254). We added in the Introduction section this sentence: Additionally, we decided to check antioxidant and pro-oxidant effects of indole-3-carbinol, because literature data do not refer to any studies that would concern such activity of indole-3-carbinol. (Lines 81-84).
Comment 4: Methods detail: The methods section lacks information on the number of biological replicates and does not clarify how final reagent concentrations were achieved, limiting reproducibility and transparency.
Response 4: We would like to thank the Reviewer for this comment. In the Subsection 2.1 we have edited the fragment as follows:
“Indole-3-carbinol, FeSO4 and H2O2 were obtained from Sigma (St. Louis, MO, USA), and concentrated C2H5OH was obtained from Stanlab (Lublin, Poland). For investigation of lipid peroxidation the LPO-586 kit was used from Enzo Life Science (Farmingdale, NY, USA). All chemicals have high level of purity. The concentrations of FeSO4 were obtained through the dilution method with use of Tris buffer (pH 7.4) and were equal to 0.0, 1200, 600, 300, 150, 75, 37.5, 18.75, 9.375, 4.687, and 2.343 µM. Concentrations of indole-3-carbinol dissolved in C2H5OH were 0.0, 20.0, 10.0, 5.0, 1.0 and 0.5 mM. The final concentration of ethanol during incubation was 1% (v/v). Hydrogen peroxide (H2O2) was used in concentration equal to 5 mM. Each experiment was performed three times. “(Lines 87-95)
Comment 5: Statistical analysis: The description of statistical methods is insufficient. It should specify which tests were applied to each dataset and indicate whether assumptions underlying ANOVA and t-tests (e.g., normality, variance homogeneity) were verified.
Response 5: We would like to thank the Reviewer for this valuable comment. In the revised manuscript, we have clarified the statistical methods applied to each dataset and included a description of the assumption checks performed prior to applying parametric tests. Specifically, we now state that for comparisons involving more than two groups, a one-way ANOVA followed by the Student–Neuman–Keuls’ post hoc test was used, whereas for pairwise comparisons (e.g., ovary vs. kidney samples), unpaired t-tests were applied. We have also added that the assumptions of normality and homogeneity of variances were tested using the Shapiro–Wilk and Levene’s tests, respectively. These clarifications have been added to the “Materials and Methods” section, and the revised text now reads:
“Statistical calculations were performed using SigmaPlot 11.0 based on data from three independent experimental replicates.
For comparisons involving more than two groups, a one-way analysis of variance (ANOVA) was performed, followed by the Student–Neuman–Keuls’ post hoc test. For pairwise comparisons (e.g., between ovary and kidney samples), unpaired t-tests were applied. Prior to applying ANOVA or t-tests, data were tested for normality using the Shapiro–Wilk test, and for homogeneity of variances using the Levene’s test. Cohen’s d effect sizes were calculated to quantify the magnitude of differences between selected treatment groups.
Statistical significance was considered at p < 0.05. All results are presented as means±SE.” (lines 120-129)
Comment 6: Results presentation: The results section is overly dense with repeated listings of concentrations and detailed data, which makes it difficult to follow. Focusing on key trends and summarizing findings more concisely would improve clarity.
Response 6: We would like to thank the Reviewer for this comment. We wanted to ensure that the description of the obtained results was accurate. Obtaining the same trend was difficult, as individual results sometimes fell outside the general conclusions that emerged during the interpretation of the graphs. However, we have edited several fragments in the Results section.
Comment 7: Statistical reporting: Statistical significance is frequently stated, but exact p-values, confidence intervals, or effect sizes are not provided. Including these metrics would strengthen the robustness and interpretability of the findings.
Response 7: We would like to thank the Reviewer for the valuable suggestion. In the revised manuscript we have addressed this point by adding exact p values to comparison where statistical significance is reported and by calculating and including Cohens d values to indicate the effect size and magnitude of differences between selected treatment groups.
Comment 8: Discussion of pro-oxidant activity: The biological implications and potential mechanisms behind the observed pro-oxidant effects of indole-3-carbinol at low FeSO₄ concentrations are not adequately explored or discussed.
Response 8: We would like to thank the Reviewer for this comment. More detailed studies would be necessary, so we would like to conduct future analyses that would allow us to examine the effect of indole-3-carbinol on the expression of selected genes, encoding proteins important for reducing the activity of reactive oxygen species. In this paper, we present the results of preliminary experiments that allowed us to determine whether indole-3-carbinol has a pro- or antioxidant effect. Moreover we evidenced that indole-3-carbinol can show pro-oxidant activity at low FeSO₄ concentrations and this is consistent with the LPO results for samples without FeSO₄ plus H2O2.
Therefore, we have added this sentence in the Result section:
“Moreover, observation of indole-3-carbinol’s dual effect is highlighted according to the data obtained for samples containing this indole derivative, but without FeSO4 and H2O2 (Figure 2).” (Lines 192-194)
Comment 9: Consistency in terminology: Throughout the manuscript, there are inconsistencies in terminology and notation (e.g., FeSO₄ vs Fe²⁺), which could confuse readers. Consistent usage should be maintained.
Response 9: We would like to thank the Reviewer for this comment. We have changed ferrous sulfate to FeSO4. The sentence is: “We compared lipid peroxidation level in the both of porcine tissues homogenates in relation to control sample and concentrations of FeSO4 (expressed as Fe2+) in each tissue homogenates.” (Lines 132-134). This sentence explains that we expressed FeSO4 as Fe2+.
Comment 10: Several sentences are overly complex or contain minor grammatical errors, which occasionally reduce clarity. A thorough language edit would enhance readability.
Response 10: We would like to thank the Reviewer for this comment. We have edited the sentences to make them grammatically correct:
- “We compared lipid peroxidation level in the both of porcine tissues homogenates in relation to control sample and concentrations of FeSO4 (expressed as Fe2+) in each tissue homogenates.” (Lines 132-134).
- We have removed “indole” and the sentence is: “Indole-3-carbinol was very fast absorbed and distributed in the cells of kidney, liver, heart, plasma, brain and lung of mice dosed with 250 mg/kg of this phytochemical and the highest concentrations of this compound were found in the kidney and liver [14].” (Lines 210-213)
- In the lines 94-95 „used” has been removed and the sentence is: “Hydrogen peroxide (H2O2) was used in concentration equal 5 mM.”
- We have edited the whole fragment in the Subsection 2.3.
“For homogenization of porcine tissues a Tris buffer (pH 7.4) was used and the reagents (FeSO4, H2O2, indole-3-carbinol) were added. Thereafter, homogenates were kept in the presence of added substances at 37°C for 30 min.
The LPO index means the level of MDA + 4-HDA (Malondialdehyde + 4-Hydroxyalkenals) and is expressed as nmol per mg of protein. The LPO-586 kit (Enzo Life Science, Farmingdale, NY, USA) was applied as a protocol in the whole described procedure. To obtain supernatant, the samples were centrifuged (5000× g, 10 min, 4°C) after incubation of homogenates with the examined substances. Thereafter, the experiment was performed in duplicate with the use of supernatant and the samples were prepared as follows: 200 μl of sample supernatant was combined with 650 μl of mixture of CH3OH and C2H3N (prepared in 1:3 ratio) containing N-methyl-2-phenylindole. Then, 150 μl of methanesulfonic acid (15.4 M) was added to each sample and the incubation was performed at 45°C for 40 min. After centrifugation (5000×g, 10 min, 4°C), an absorbance of each sample was measured at 586 nm. An absorbance of protein samples was measured at 595 nm. Results are expressed as MDA + 4-HDA per 1 mg of protein.” (Lines 104-118)
Comment 11: Ethical considerations: The manuscript does not include any statement regarding ethical approval or adherence to animal use guidelines for the collection of porcine tissues, which is important even when using post-mortem sample
Response 11: We would like to thank the Reviewer for this comment. According to The Polish Act on the Protection of Animals Used for Scientific or Educational Purpose and Directive 2010/63/EU of the European Parliament and the Council (published on 22 September 2010) it is not necessary to have an approval of the local ethics committee. It should be stressed that we did not use experimental animals in the current study.
The fragment of the Subsection 2.2. after the change is: “According to The Polish Act on the Protection of Animals Used for Scientific or Educational Purpose and Directive 2010/63/EU of the European Parliament and the Council (published on 22 September 2010) it is not necessary to have an approval of the local ethics committee. Kidneys and ovaries tissues were obtained from pigs (aged 8-9 months) at a slaughterhouse. The animals were in good health. The tissues were immediately frozen on solid CO2 and stored at −80 °C.” (Lines 97-102)
Overall, addressing these points will improve the manuscript’s clarity, methodological rigor, and contribution to the literature on sugar intake reduction strategies.
We would like to thank the Reviewer for this comment.
Reviewer 2 Report
Comments and Suggestions for Authors
The authors offer their work detailing the two faces of I3C as a dietary supplement noting their analysis of lipid peroxidation by Fenton reaction. They noted that I3C has dose-dependent antioxidant effects in porcine ovary and kidney homogenates under high oxidative stress, reducing Fenton reaction-induced lipid peroxidation.
Overall this is a straightforward and well performed study. The authors used dose dependent ferrous sulfation and hydrogen peroxide as their destructive agents. I think it would improve the study to use one or two more destructive agents, perhaps DMSO to show that the I3C effect can be replicated at least in triplicate to show that I3C is truly making the difference in oxidative stress.
Author Response
The authors offer their work detailing the two faces of I3C as a dietary supplement noting their analysis of lipid peroxidation by Fenton reaction. They noted that I3C has dose-dependent antioxidant effects in porcine ovary and kidney homogenates under high oxidative stress, reducing Fenton reaction-induced lipid peroxidation.
We would like to thank the Reviewer for this positive comment.
Overall this is a straightforward and well performed study. The authors used dose dependent ferrous sulfation and hydrogen peroxide as their destructive agents. I think it would improve the study to use one or two more destructive agents, perhaps DMSO to show that the I3C effect can be replicated at least in triplicate to show that I3C is truly making the difference in oxidative stress.
We would like to thank the Reviewer for this comment. We will consider using DMSO or other prooxidants as additional destructive agents in the future studies.
Round 2
Reviewer 1 Report
Comments and Suggestions for Authors
No more comments